# Dielectric and Spin-Glass Magnetic Properties of the A-Site Columnar-Ordered Quadruple Perovskite Sm_2_CuMn(MnTi_3_)O_12_

**DOI:** 10.3390/ma15238306

**Published:** 2022-11-23

**Authors:** Alexei A. Belik, Ran Liu, Kazunari Yamaura

**Affiliations:** 1International Center for Materials Nanoarchitectonics (WPI-MANA), National Institute for Materials Science (NIMS), Namiki 1-1, Tsukuba 305-0044, Ibaraki, Japan; 2Graduate School of Chemical Sciences and Engineering, Hokkaido University, North 10 West 8, Kita-ku, Sapporo 060-0810, Hokkaido, Japan; 3Institute of Scientific and Industrial Research, Osaka University, Mihogaoka 8-1, Ibaraki 567-0047, Osaka, Japan

**Keywords:** quadruple perovskites, A-site columnar-ordered, antisite disorder, crystal structures, spin-glass

## Abstract

Perovskite-type ABO_3_ oxides show a number of cation-ordered structures, which have significant effects on their properties. The rock-salt-type order is dominant for B cations, and the layered order for A cations. In this work, we prepared a new perovskite-type oxide, Sm_2_CuMn(MnTi_3_)O_12_, with a rare columnar A-site order using a high-pressure, high-temperature method at about 6 GPa and about 1700 K. Its crystal structure was studied with synchrotron powder X-ray diffraction. The compound crystallizes in space group *P*4_2_/*nmc* (No. 137) at room temperature with *a* = 7.53477 Å and *c* = 7.69788 Å. The magnetic properties of the compound were studied with dc and ac magnetic susceptibility measurements and specific heat. Spin-glass (SG) magnetic properties were found with *T*_SG_ = 7 K, while specific heat, in the form of *C*_p_/*T*, showed a strong, very broad anomaly developing below 20 K and peaking at 4 K. The dielectric constant of Sm_2_CuMn(MnTi_3_)O_12_ was nearly frequency and temperature independent between 8 K and 200 K, with a value of about 50. Cu^2+^ doping drastically modified the magnetic and dielectric properties of Sm_2_CuMn(MnTi_3_)O_12_ in comparison with the parent compound Sm_2_MnMn(MnTi_3_)O_12_, which showed a long-range ferrimagnetic order at 34–40 K. The antisite disorder of Cu^2+^ and Mn^2+^ cations between square-planar and octahedral sites was responsible for the SG magnetic properties of Sm_2_CuMn(MnTi_3_)O_12_.

## 1. Introduction

The properties of the perovskite-structure oxide material, ABO_3_, are controlled by their chemical compositions and degrees of cation orderings [1,2]. There are perovskites with B-site cation orderings, A-site cation orderings, and both types of orderings. In the case of B-site ordering, the rock-salt-type order is dominant [3]. In the case of A-site ordering in ABO_3_, the layered-type order is dominant [1,2,4], but there are other types of ordering [5,6]. There are also two special families of perovskites with A-site orderings: A-site-ordered quadruple perovskites, AA′_3_B_4_O_12_ [7,8,9], and A-site columnar-ordered quadruple perovskites, A_2_A′A″B_4_O_12_ [10]. Quadruple perovskites can have ordered arrangements of 3*d* transition metals at the A (in general) perovskite sites in addition to the B sites. The resulting B–B, A–B, and A–A exchange interactions can produce complex interaction patterns and frustration networks and result in competing magnetic ground states, a large number of magnetic transitions and unexpected magnetism [11].

With A = R = rare earth elements and Bi and A′ = A″ = B = Mn, interesting classes of perovskite manganites are formed, namely RMn_7_O_12_ [9] and RMn_3_O_6_ (in a short formula) [12]. They show several magnetic transitions with spin reorientations [9], and some members have incommensurate magnetic structures [13]. The Cu^2+^ doping of RMn_7_O_12_ and RMn_3_O_6_ has beneficial effects on their magnetic properties. in a sense that magnetic transition temperatures significantly increase, e.g., from about 80–87 K in RMn_7_O_12_ [13] to 360–400 K in RCu_3_Mn_3_O_12_ [14], and from about 60–77 K in RMn_3_O_6_ [12] to 160–180 K in R_2_CuMnMn_4_O_12_ [11,15]. In case of BiMn_7_O_12_ [9], Cu^2+^ doping results in complex structural behavior [16,17,18], complex magnetic behavior, and almost a linear rise of magnetic transition temperatures in BiCu*_x_*Mn_7–*x*_O_12_ [18] for 0.8 ≤ *x* ≤ 3 from about 30 K (for 0 < *x* < 0.8) to 360 K for *x* = 3 [19]. The beneficial effects of Cu^2+^ doping also took place in Y_2_MnGaMn_4_O_12_, which shows spin-glass magnetic properties at 26 K [20], as Y_2_CuGaMn_4_O_12_ exhibits long-range ferrimagnetic ordering at 115 K [21].

Sm_2_MnMn(MnTi_3_)O_12_ is a member of the A-site columnar-ordered quadruple perovskites. The magnetic properties of Sm_2_MnMn(MnTi_3_)O_12_ [22,23] were somewhat unexpected, as it shows a long-range ferrimagnetic ordering with *T*_C_ = 34–40 K and a well-defined M–H hysteresis loop with remnant magnetization of 2.3–2.4 μ_B_/f.u. at 5 K. The concentration of 3*d* transition metals (Mn^2+^) at the B sites (25%) was below the percolation limit for the corner-shared octahedral network. Nevertheless, Mn^2+^ cations at the B sites were involved in the long-range ordering with a noticeable ordered magnetic moment [23]. Similar compounds without magnetic cations at the B sites (e.g., Ca_2_MnMnTi_4_O_12_ [24] and NaRMnMnTi_4_O_12_ [25]) show antiferromagnetic (AFM) transitions at lower temperatures of about 12 K. Therefore, there should be a noticeable involvement of the A–A and A–B exchange interactions in Sm_2_MnMn(MnTi_3_)O_12_ to stabilize the ferrimagnetic structure out of AFM one and to increase the magnetic transition temperature. In addition, Sm_2_MnMn(MnTi_3_)O_12_ was the first example among A-site columnar-ordered quadruple perovskites, demonstrating relaxor-type dielectric properties with broad maxima on the temperature dependence of a dielectric constant near 220 K [22].

In this work, we investigated the effects of Cu^2+^ doping on the magnetic and dielectric properties of the parent Sm_2_MnMn(MnTi_3_)O_12_ compound and prepared Sm_2_CuMn(MnTi_3_)O_12_ using a high-pressure, high-temperature method. However, in this case, the magnetic properties of the parent compound were “degraded” by Cu^2+^ doping, as only spin-glass (SG) magnetic properties were observed below *T*_SG_ = 7 K. We attributed this degradation to antisite disorder. The relaxor-type dielectric properties of the parent compound disappeared, and Sm_2_CuMn(MnTi_3_)O_12_ demonstrated a frequency and temperature independent dielectric constant between 10 K and 200 K, with a value of about 50.

## 2. Experimental

Sm_2_CuMn(MnTi_3_)O_12_ was prepared using a high-pressure, high-temperature method using a belt-type high-pressure machine at 6 GPa and about 1700 K for 2 h in a Pt capsule. After annealing at 1700 K, the samples were quenched to room temperature (RT) by turning off the heating current, and the pressure was slowly released. Stoichiometric amounts of Sm_2_O_3_ (99.9%), CuO (99.9%), MnO (99.99%), and TiO_2_ (99.9%) were used as an initial oxide mixture with the 1:1:2:3 ratio, respectively. Commercial Sm_2_O_3_, CuO, and TiO_2_ chemicals were used. A single-phase MnO oxide was prepared from a commercial MnO_2_ chemical by annealing at 1273 K for 4 h in a 20% H_2_ + 80% Ar gas flow.

X-ray powder diffraction (XRPD) data were collected at RT with a RIGAKU MiniFlex600 diffractometer (CuKα radiation; a 2*θ* range of 8–100°; a step of 0.02°, and scan speed of 1°/min). The synchrotron XRPD data were collected at RT on the BL15XU beamline (the former NIMS beamline) of SPring-8 [26] between 2.04° and 60.23° at 0.003° intervals in 2θ with a wavelength of *λ* = 0.65298 Å. The data between 6° and 60.23° were used in the refinements as no reflections were observed and expected below 6°. The sample was inserted into a Lindemann glass capillary tube (inner diameter: 0.1 mm), which was rotated during the measurements. The Rietveld analysis of all XRPD data was performed using the *RIETAN-2000* program [27].

Scanning electron microscopy (SEM) images and energy-dispersive X-ray (EDX) spectra were obtained on a Hitachi Miniscope TM3000 (operating at 15 kV).

SQUID magnetometers (Quantum Design, MPMS-XL-7T and MPMS3) were used for the magnetic measurements. Temperature dependence was measured between 2 and 400 K in applied fields of 100 Oe and 10 kOe under both zero-field-cooled (ZFC) and field-cooled on cooling (FCC) conditions on an MPMS-XL-7T. Magnetic-field dependence was measured at *T* = 2 K and 5 K between −70 and 70 kOe on MPMS3. Frequency dependent alternating current (ac) susceptibility measurements were performed on cooling with a Quantum Design MPMS3 instrument at different frequencies (*f*), different applied oscillating magnetic fields (*H*_ac_), and different static dc field (*H*_dc_). Relaxation curves were measured on MPMS3 using the following procedure: the sample was cooled down from 50 K to a measurement temperature at zero magnetic field, then a magnetic field of 100 Oe was applied, and magnetization was measured (as one scan within 2 s) as a function of time every 5 s.

Specific heat, *C*_p_, was measured by cooling from 270 K to 2 K at zero magnetic field and from 150 K to 2 K at magnetic field of 90 kOe by a pulse relaxation method using a commercial calorimeter (Quantum Design PPMS).

The dielectric constant and dielectric loss were measured on a NOVOCONTROL Alpha-A High Performance Frequency Analyzer in a frequency range from 100 Hz to 665 kHz in a temperature range from 8 K to 330 K (on heating) at zero magnetic field.

## 3. Results and Discussion

The as-synthesized Sm_2_CuMn(MnTi_3_)O_12_ contained a small amount of CuO impurity. In addition, the synchrotron XRPD pattern showed the presence of Pt impurity. However, Pt appeared from Pt capsules used in the synthesis and can be considered as an extrinsic impurity. The presence of a CuO impurity suggests that the main phase should be slightly Cu-deficient in comparison with the target composition. The morphology of the sample is shown in Figure 1. The particle sizes varied between about 10 and 50 μm. The Ti:Mn:Sm:Cu cation ratio determined with EDX was 3.24(8):2.02(5):1.96(4):0.77(7), respectively. These values were close to the nominal values within 3σ.

All of the reflections on the laboratory and synchrotron XRPD patterns (except CuO and Pt) could be indexed in a tetragonal system in space group *P*4_2_/*nmc* (No. 137) (Figure 2). Sm_2_CuMnMnTi_3_O_12_ was found to crystallize in the parent structure of the A-site columnar-ordered quadruple perovskites, A_2_A′A″B_4_O_12_ [10]. Therefore, the structural data for the parent compound Sm_2_MnMn(MnTi_3_)O_12_ [22,23] were taken as an initial starting model.

In the structural analysis, we first assumed ideal cation distributions (that is, Sm at the A site, Cu at the square-planar A′ site, Mn at the tetrahedral A″ site, and 0.75Ti + 0.25Mn at the octahedral B site) and refined the occupation factors (*g*) together with all of the other structural and nonstructural parameters (except *g*(B): one cation occupation factor should always be fixed to avoid significant correlations among the refined *g* parameters). In addition, in the structural analysis, we always assumed that Ti^4+^ cations were located at the B site, as Ti^4+^ cations strongly prefer octahedral sites [28].

The refined *g* values were as follows: *g*(Sm–A) = 0.9167(16), *g*(Cu–A′) = 0.914(7), and *g*(Mn–A″) = 1.047(7). These values suggest that the ideal cation distribution was not realized, and there were some antisite disorders. The *g*(Cu–A′) value suggested that this site should contain lighter elements that could only be Mn (with the above assumption on Ti). When only Mn was placed at the square-planar A′ site, the occupation factor was *g*(Mn–A′) = 1.112(8), meaning that heavier elements should also be at this site. Because it was difficult to precisely refine the distribution of Mn and Cu with X-ray diffraction, we introduced a virtual atom: MC = 0.5Mn + 0.5Cu. The precise distribution of Mn and Cu could only be determined with neutron diffraction. The disordering of cations at the Cu site was also observed as in many cases of such perovskites [21,22,25,29].

The refined structural parameters and primary bond lengths and angles in Sm_2_CuMn(MnTi_3_)O_12_ are listed in Table 1 and Table 2. The experimental, calculated, and difference synchrotron patterns are shown in Figure 3. The crystal structure of Sm_2_CuMn(MnTi_3_)O_12_ is illustrated in the inset of Figure 3.

Our model suggested that a small fraction of Cu^2+^ cations should be located at the B site. Indirect evidence for the location of Cu^2+^ cations at the octahedral site can be seen from the resulting Ti/MC–O bond lengths. In the parent and related compounds, R_2_MnMn(MnTi_3_)O_12_ (R = Nd, Sm, Eu, and Gd), the Ti/Mn–O bond lengths were about 1.99, 2.01, and 2.01 Å (from both the synchrotron [22,29] and neutron [23] diffraction data), resulting in an octahedral distortion parameter, Δ, of about 0.2 × 10^–4^. On the other hand, in Sm_2_CuMn(MnTi_3_)O_12_, the Ti/MC–O bond lengths were about 1.95, 1.99, and 2.02 Å resulting in Δ of about 2.0 × 10^–4^. This rise in the octahedral distortion could be caused by the presence of a small amount of Jahn–Teller active Cu^2+^ cations at this site.

Magnetic susceptibility curves, *χ* versus *T*, of Sm_2_CuMn(MnTi_3_)O_12_ under applied magnetic fields of 0.1 kOe and 10 kOe are shown on Figure 4. There was a divergence between the 100 Oe ZFC and FCC curves at 7 K and a relatively sharp maximum on the 100 Oe ZFC curve at 7 K. A divergence between the ZFC and FCC curves almost disappeared under 10 kOe. These features are typical for spin-glass transitions [30,31,32]. Isothermal magnetization, *M* versus *H*, curves demonstrated an extended S-type shape with very weak and narrow hysteresis (Figure 5). Almost no hysteresis was observed at 5 K because 5 K was close to its *T*_SG_ = 7 K; on the other hand, the hysteresis was noticeably wider at a lower temperature of 2 K. Such *M* versus *H* curves are also typical for spin glasses [30,31,32].

The inverse magnetic susceptibilities (*χ*^−1^ versus *T*) followed the Curie–Weiss law at high temperatures (Figure 4). To obtain the effective magnetic moment and the Curie–Weiss temperature, we performed fits between 250 and 345 K using the 10 kOe FCC curves (the fit and fitting parameters are summarized on Figure 4). The experimental effective magnetic moment was close to the expected one (8.803 μ_B_; in the calculations we used 1.5 μ_B_ for Sm^3+^ [33]). The negative Curie–Weiss temperature shows that the main magnetic interactions were antiferromagnetic in nature. The ratio between the Curie–Weiss temperature (−81.5 K) and *T*_SG_ (the so-called frustration ratio) was about 11, indicating a strong degree of magnetic frustration. We note that CuO was in an antiferromagnet with transition temperatures of 213 K and 230 K. Therefore, CuO impurity should not affect the reported magnetic properties at low temperatures.

To confirm the spin-glass nature of the sample, we measured ac magnetic susceptibility curves (Figure 6 and Figure 7). We note that no dependence of the χ′ and χ″ values on the applied *H*_ac_ field was observed (inset of Figure 6). We indeed observed typical features of spin-glasses: peak positions were frequency-dependent and shifted to higher temperatures with increasing frequency; in addition, peak intensity was suppressed on the χ′ versus *T* curves and enhanced on the χ″ versus *T* curves with increasing frequency. All of these features are typical for spin glasses [30,31,32]. In addition, the shape of the χ′ versus *T* and the χ″ versus *T* curves was also typical for spin glasses. The criterion, which quantifies the relative change of the spin-glass temperature per frequency decade and is defined as Δ*T*_SG_/[*T*_SG_Δlog(*f*)], was about 0.023 for Sm_2_CuMn(MnTi_3_)O_12_ (with *T*_SG_ = 7.2 K at *f* = 2 Hz and *T*_SG_ = 7.6 K at *f* = 500 Hz). This value is often observed in different spin-glass materials [30,31,32].

Sm_2_CuMn(MnTi_3_)O_12_ shows time-dependent magnetic properties below *T*_SG_, namely magnetization relaxation (Figure 8). Above *T*_SG_, no noticeable relaxation of magnetization was detected. Time-dependent magnetic properties, such as relaxation, are typical features of spin-glass systems. Relaxation below *T*_SG_ was fitted by the stretched exponential function, *f*(*t*) = *M*_0_ − *M*_SG_ × exp[−(*t*/*t_r_*)*^β^*] [30], and the resultant parameters are listed on Table 3. The most important parameter is the mean relaxation time, *t_r_*, and it decreases monotonically with increasing temperature.

The specific heat data showed a noticeable magnetic contribution to the total specific heat below about 20 K, where it could be clearly seen as a rise in *C*_p_/*T* values below 20 K (Figure 9). No λ-type anomaly was detected in the *C*_p_ versus *T* curve (inset of Figure 9, a green curve). Instead, a broad anomaly was seen in the *C*_p_ versus *T* curve, which gave a broad peak centered at 4 K in the *C*_p_/*T* versus *T* curve. Therefore, specific heat measurements confirmed the absence of long-range magnetic ordering. A magnetic field of 90 kOe slightly suppressed the peak near 4 K and moved the magnetic entropy into the 14–40 K range.

The temperature dependence of the dielectric constant and dielectric loss is shown in Figure 10. The dielectric constant was nearly temperature and frequency-independent between 8 and 200 K. Above about 200 K, a sharp rise in the dielectric constant was observed, where the magnitude of the rise depended on frequency. This behavior typically originates from the Maxwell–Wagner contribution due to increased conductivity. No broad anomalies were observed in Sm_2_CuMn(MnTi_3_)O_12_ in comparison with the parent compound Sm_2_MnMn(MnTi_3_)O_12_. This fact shows that Cu^2+^ doping drastically modified the dielectric properties as well, in addition to the magnetic properties. We note that Pt impurity was only observed in a powder sample, which could contain parts from the surface. The surfaces of a pellet used for dielectric measurements were polished. Therefore, Pt impurity should not present in a pellet and affect dielectric measurements.

Spin-glass magnetic properties were also observed in Sm_2_MnZn(MnTi_3_)O_12_ at *T*_SG_ = 6.5 K, with a significant antisite disorder [34]. This fact shows that antisite structural disorder should play a major role in the modification of magnetic properties of the parent Sm_2_MnMn(MnTi_3_)O_12_ compound, not the nature of dopant cations (magnetic as Cu^2+^ or non-magnetic as Zn^2+^). Both Sm_2_CuMn(MnTi_3_)O_12_ and Sm_2_MnZn(MnTi_3_)O_12_ demonstrated similar low-temperature specific heat features (Figure 9b).

The beneficial effects of Cu^2+^ doping in RMn_7_O_12_ [13,14], RMn_3_O_6_ [11,12,15], and Y_2_MnGaMn_4_O_12_ [20,21] originate from the fact that Cu^2+^ doping is aliovalent doping, which produces Mn^4+^ cations. A mixture of Mn^3+^ and Mn^4+^ at the B sites of perovskites significantly enhanced the exchange interactions and magnetic transition temperatures. On the other hand, Cu^2+^ doping in the parent Sm_2_MnMn(MnTi_3_)O_12_ compound was isovalent doping. Such doping did not change the oxidation state of Mn, while the antisite disordering “degraded” the magnetic properties.

## 4. Conclusions

A new member of the A-site columnar-ordered quadruple perovskite family, Sm_2_CuMn(MnTi_3_)O_12_, was prepared using a high-pressure, high-temperature method. Cu^2+^ doping significantly modified the properties of the parent Sm_2_MnMn(MnTi_3_)O_12_ compound, as spin-glass magnetic properties at *T*_SG_ = 7 K were observed in Sm_2_CuMn(MnTi_3_)O_12_ in comparison with the long-range ferrimagnetic order at *T*_C_ = 34–40 K in Sm_2_MnMn(MnTi_3_)O_12_. In addition, relaxor-like dielectric properties of Sm_2_MnMn(MnTi_3_)O_12_ disappeared in Sm_2_CuMn(MnTi_3_)O_12_, which showed a nearly temperature and frequency-independent dielectric constant between 8 and 200 K with a value of about 50.

## Figures and Tables

**Figure 1 materials-15-08306-f001:**
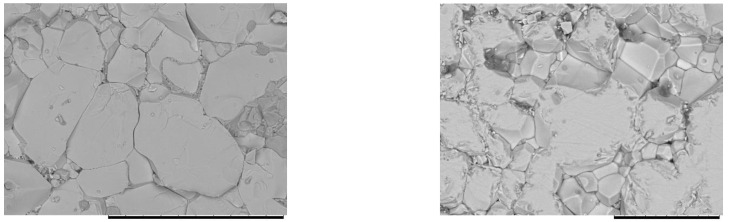
Scanning electron microscopy (SEM) images of the fractured surface of the as-synthesized Sm_2_CuMn(MnTi_3_)O_12_ sample. The scale bars are 100 µm (**left**) and 30 µm (**right**); magnification is 1000 (**left**) and 2000 (**right**). The surface is partially polished in the right panel. No polishing has been done in the left panel.

**Figure 2 materials-15-08306-f002:**
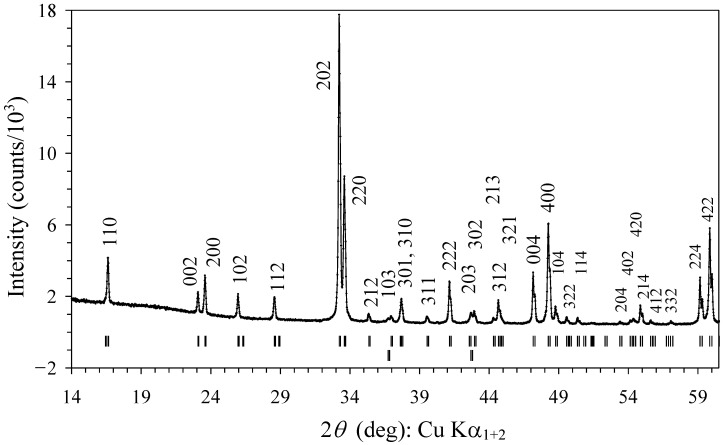
Laboratory powder X-ray diffraction pattern of Sm_2_CuMn(MnTi_3_)O_12_ in a 2θ range from 14° to 60.5°. Possible Bragg reflection positions for Sm_2_CuMn(MnTi_3_)O_12_ (the first row) and CuO impurities (the second row) are shown. The (*hkl*) indices of all of the observed reflections of Sm_2_CuMn(MnTi_3_)O_12_ are given.

**Figure 3 materials-15-08306-f003:**
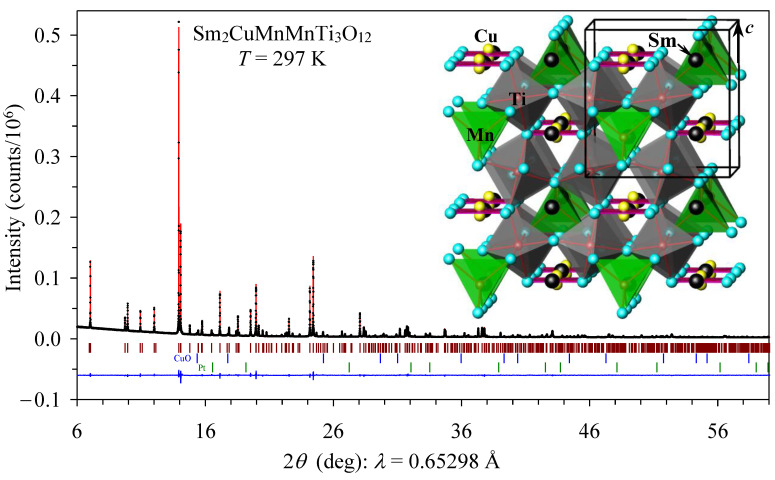
Full experimental (black crosses), calculated (red line), and difference (blue line at the bottom) room-temperature synchrotron powder X-ray diffraction patterns of Sm_2_CuMn(MnTi_3_)O_12_ in a 2*θ* range of 6° and 60°. The brown tick marks show possible Bragg reflection positions for the main phase, the blue tick marks are for CuO impurity (2.0 wt.%), and the green ones are for Pt impurity (0.3 wt.%). The inset shows a tetragonal crystal structure of Sm_2_CuMn(MnTi_3_)O_12_; TiO_6_ octahedra (gray), MnO_4_ tetrahedra (green), and ideal CuO_4_ square-planar units (red) are plotted; Sm atoms are given by black circles; split Cu sites are shown by yellow circles.

**Figure 4 materials-15-08306-f004:**
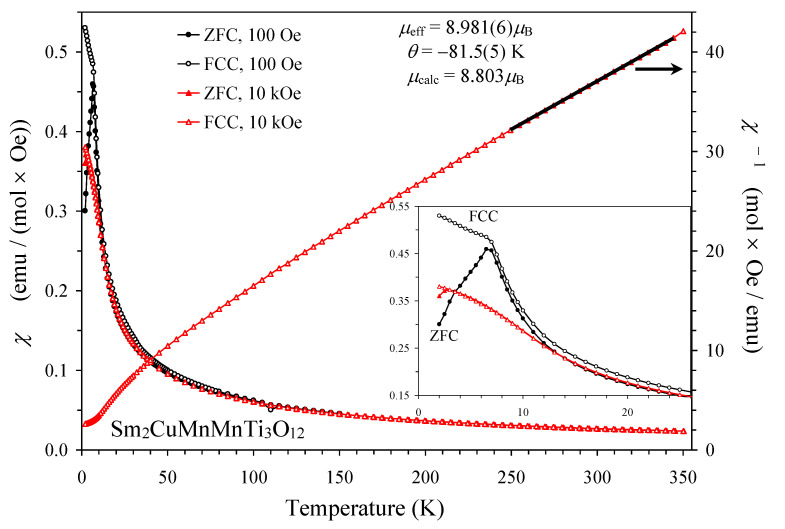
The left-hand axis shows the ZFC (filled symbols) and FCC (empty symbols) dc magnetic susceptibility curves (*χ* = *M*/*H*) of Sm_2_CuMn(MnTi_3_)O_12_ at 100 Oe (black) and 10 kOe (red). The right-hand axis gives the 10 kOe FCC *χ*^−1^ versus *T* curve with the Curie–Weiss fit between 250 K and 345 K (black line). Parameters of the fits are shown on the figure. The inset shows details below 30 K.

**Figure 5 materials-15-08306-f005:**
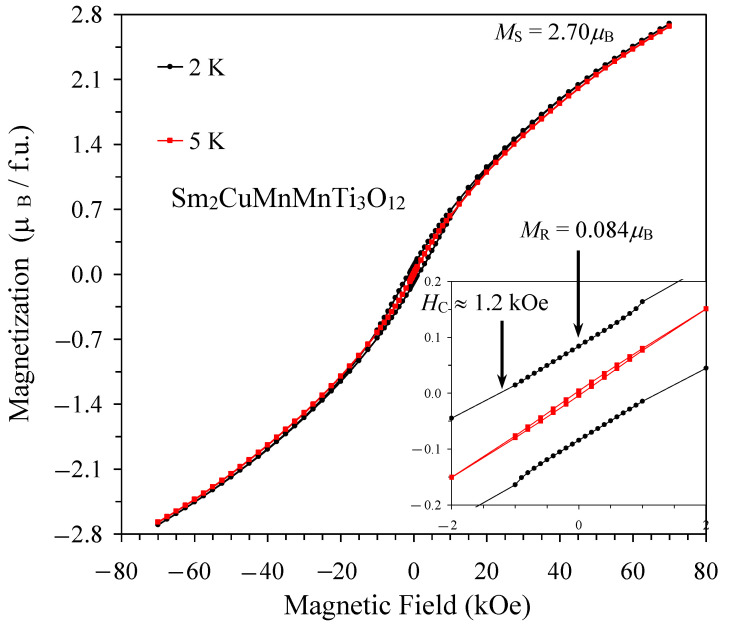
*M* versus *H* curves of Sm_2_CuMn(MnTi_3_)O_12_ at *T* = 2 K (black) and *T* = 5 K (red) (f.u.: formula unit). The inset shows details near the origin. Parameters of the *M* versus *H* curve at *T* = 2 K are given: *M*_S_ is the magnetization value at *H* = 70 kOe, *M*_R_ is the remnant magnetization, and *H*_C_ is the coercive field.

**Figure 6 materials-15-08306-f006:**
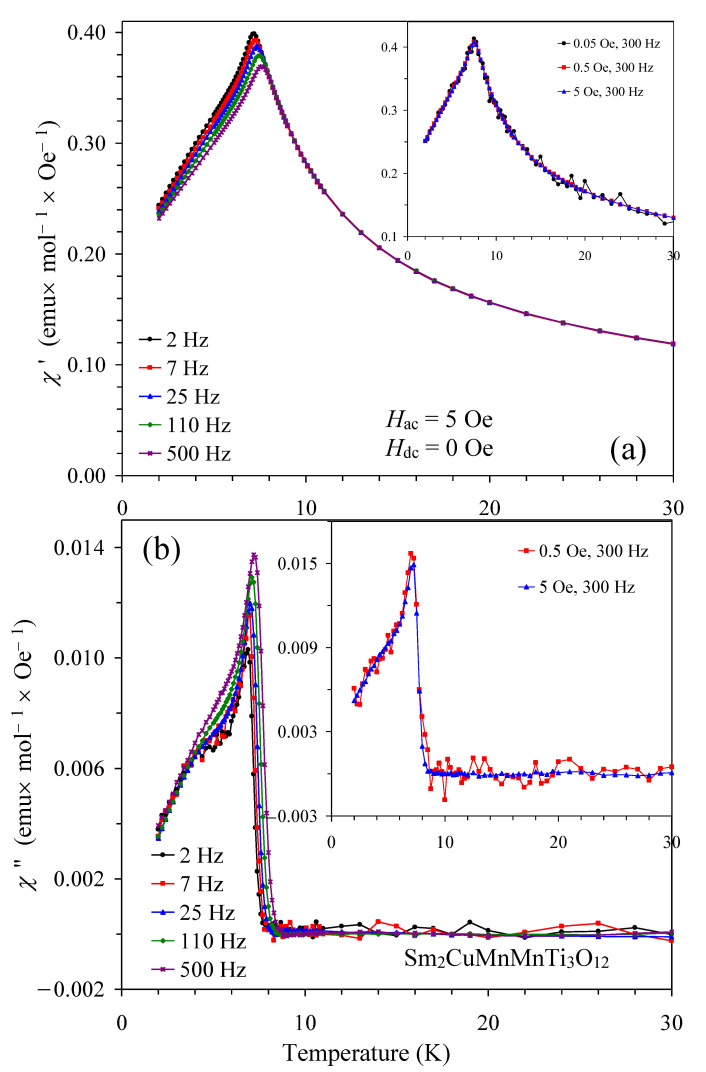
(**a**) Real (χ′) and (**b**) imaginary (χ″) parts of the ac magnetic susceptibility curves of Sm_2_CuMn(MnTi_3_)O_12_ at different frequencies. The insets in (**a**,**b**) show the χ′ versus *T* and χ″ versus *T* curves at different *H*_ac_ = 0.05, 0.5, and 5 Oe and one frequency (*f* = 300 Hz) (the χ″ data at *H*_ac_ = 0.05 Oe are not shown because they were too noisy).

**Figure 7 materials-15-08306-f007:**
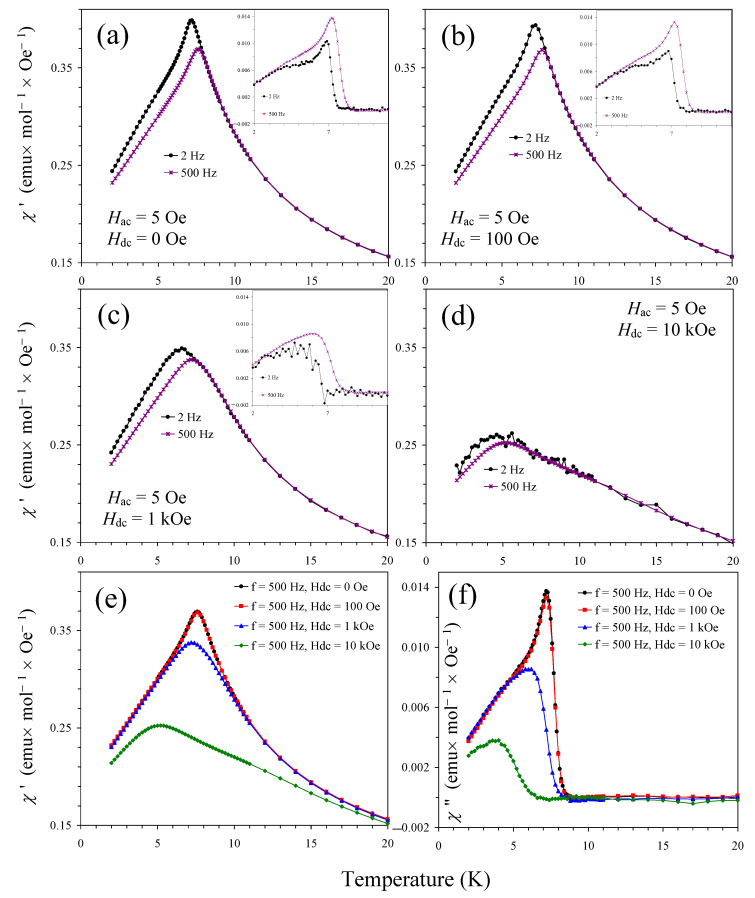
(**a**–**d**) χ′ versus *T* curves of Sm_2_CuMn(MnTi_3_)O_12_ at *f* = 2 Hz and 500 Hz and different bias dc fields: (**a**) *H*_dc_ = 0 Oe, (**b**) 100 Oe, (**c**) 1 kOe, and (**d**) 10 kOe. Insets show χ″ versus *T* curves (the χ″ data at *f* = 2 Hz and *H*_dc_ = 10 kOe were too noisy and not shown). (**e**) All χ′ versus *T* curves at *f* = 500 Hz are shown in one figure. (**f**) All χ″ versus *T* curves at *f* = 500 Hz are shown in one figure.

**Figure 8 materials-15-08306-f008:**
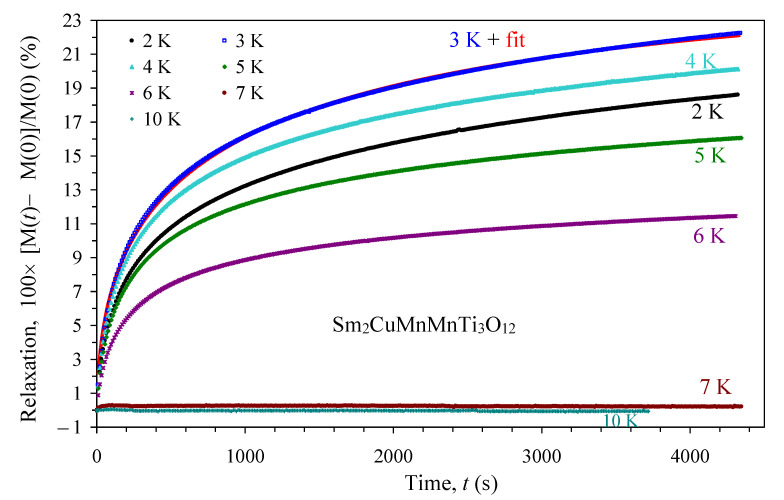
Relaxation curves defined as 100 × [M(*t*) − M(0)]/M(0) versus time (*t*) for Sm_2_CuMn(MnTi_3_)O_12_ at temperatures of 2, 3, 4, 5, 6, 7, and 10 K. Experimental points are given by symbols, and the red line shows the fit at 3 K as an example. The equation used for fitting and the resultant parameters are listed in Table 3.

**Figure 9 materials-15-08306-f009:**
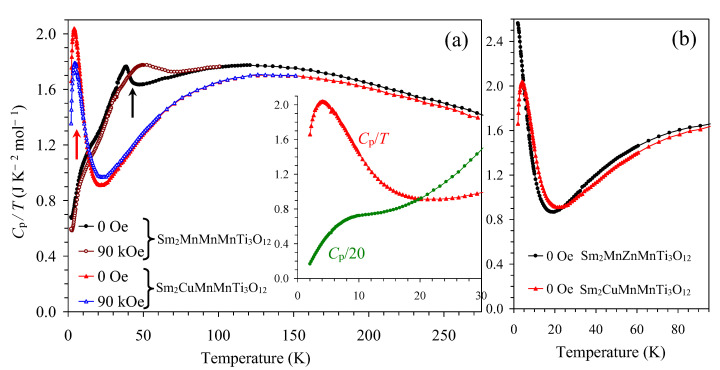
(**a**) *C*_p_/*T* versus *T* curves of Sm_2_CuMn(MnTi_3_)O_12_ at *H* = 0 Oe (red triangles) and 90 kOe (blue triangles) in comparison with the parent compound Sm_2_MnMn(MnTi_3_)O_12_ at *H* = 0 Oe (black circles) and 90 kOe (brown circles). *C*_p_ is the total specific heat. The arrows show the positions of the magnetic anomalies. The inset shows the *C*_p_/*T* versus *T* curve at *H* = 0 Oe below 30 K, and the *C*_p_ versus *T* curve at *H* = 0 Oe (green circles). For the *C*_p_ versus *T* curve, the *C*_p_ values were divided by 20, and the *C*_p_ unit is J K^−1^ mol^−1^. (**b**) Comparison of *C*_p_/*T* versus *T* data for Sm_2_CuMn(MnTi_3_)O_12_ (red triangles) and Sm_2_MnZn(MnTi_3_)O_12_ (black circles) [34] at *H* = 0 Oe.

**Figure 10 materials-15-08306-f010:**
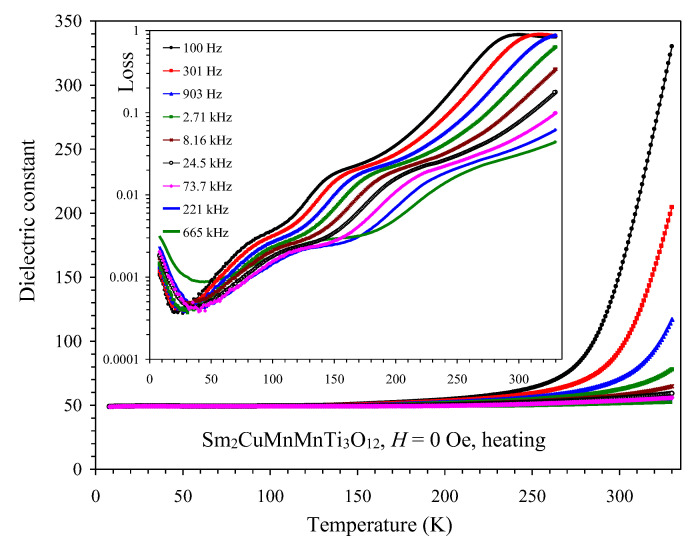
Temperature dependence of the dielectric constant at different frequencies in Sm_2_CuMn(MnTi_3_)O_12_. The inset shows temperature dependence of dielectric loss in the logarithmic scale.

**Table 1 materials-15-08306-t001:** Structure parameters of Sm_2_CuMn(MnTi_3_)O_12_ from synchrotron powder diffraction data (*λ* = 0.65298 Å) at room temperature.

Crystal system	Tetragonal
Space group	*P*4_2_/*nmc* (No. 137, cell choice 2)
*Z*	2
Caclulated density (g/cm^3^)	6.15
Formula weight (g/cm^3^)	809.737
Used *d* range (Å)	0.6507–6.238
*a* (Å)	7.53477(1)
*c* (Å)	7.69788(1)
*V* (Å^3^)	437.0301(8)
*g*(Sm)	0.9413(9)Sm + 0.0587Mn
*z*(Sm)	0.22194(4)
*B*(Sm) (Å^2^)	0.796(7)
*g*(Cu)	0.4597MC + 0.0403Sm
*z*(Cu)	0.7804(3)
*B*(Cu) (Å^2^)	1.25(7)
*g*(Mn)	0.963(3)Mn + 0.037Sm
*B*(Mn) (Å^2^)	0.38(6)
*g*(Ti)	0.25MC + 0.75Ti
*B*(Ti) (Å^2^)	0.400(9)
*y*(O1)	0.0571(3)
*z*(O1)	−0.0379(3)
*B*(O1) (Å^2^)	0.33(5)
*y*(O2)	0.5363(3)
*z*(O2)	0.5745(3)
*B*(O2) (Å^2^)	0.36(5)
*x*(O3)	0.44161(23)
*B*(O3) (Å^2^)	1.48(7)
*R*_wp_ (%)	3.16
*R*_p_ (%)	2.17
*R*_I_ (%)	2.63
*R*_F_ (%)	1.69

The Sm site is in the 4*d* site (0.25, 0.25, *z*); Cu is in the 4*c* site (0.75, 0.25, *z*); Mn is in the 2*b* site (0.75, 0.25, 0.25); Ti is in the 8*e* site (0, 0, 0); O1 and O2 are in the 8*g* site (0.25, *y*, *z*), and O3 is in the 8*f* site (*x*, −*x*, 0.25). *g* is the occupation factor. *g*(O1) = 1, *g*(O2) = 1, and *g*(O3) = 1. MC is a virtual atom: 0.5Mn + 0.5Cu.

**Table 2 materials-15-08306-t002:** Bond lengths (in Å), bond angles (in deg), and distortion parameters of TiO_6_ (Δ) in Sm_2_CuMn(MnTi_3_)O_12_ at room temperature.

Sm–O1 × 2	2.352(3)
Sm–O1 × 2	2.472(2)
Sm–O2 × 2	2.437(2)
Sm–O3 × 4	2.744(1)
Cu–O3 × 4	2.055(2)
Mn–O2 × 4	2.102(3)
Ti–O1 × 2	1.954(1)
Ti–O2 × 2	1.988(1)
Ti–O3 × 2	2.023(1)
Δ(TiO_6_)	2.0 × 10^−4^
Ti–O1–Ti × 2	149.17(9)
Ti–O2–Ti × 2	142.75(9)
Ti–O3–Ti × 2	144.17(9)

**Table 3 materials-15-08306-t003:** Results of the fittings of the relaxation curves of Sm_2_CuMn(MnTi_3_)O_12_ at different temperatures.

*T* (K)	*M* _0_	*M* _SG_	*t_r_* (s)	*β*
2	22.80(13)	23.67(18)	1270(24)	0.4362(4)
3	26.01(12)	27.20(20)	979(16)	0.4461(4)
4	22.81(10)	23.94(18)	800(11)	0.4495(4)
5	17.74(6)	18.62(13)	671(8)	0.4558(4)
6	12.34(14)	13.01(9)	554(6)	0.4654(4)

The fitting equation is *f*(*t*) = *M*_0_ − *M*_SG_ × exp[−(*t*/*t_r_*)*^β^*] [30] applied to the 100 × [M(*t*) − M(0)]/M(0) versus time (*t*) curves.

## Data Availability

Data are available from A.A.B. upon reasonable request.

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
