# Peer review of "Dielectric and Spin-Glass Magnetic Properties of the A-Site Columnar-Ordered Quadruple Perovskite Sm2CuMn(MnTi3)O12"

_materials, 2022, doi:10.3390/ma15238306_

Round 1

Reviewer 1 Report

Given in the attached file.

Author Response

Reviewer 1.

  1. In the experimental section the authors are suggested to mention the exact stoichiometric composition/molar ratio in digits and the quality of each starting material for synthesizing their QP compound. Several references of QPs/double perovskites can be cited in the introduction for examples (1) https://doi.org/10.1002/ange.201400607 (2) https://doi.org/10.1021/acs.chemmater.5b02386 (3) https://doi.org/10.1038/nature07816 to improve the manuscript and further address similar structural materials.

Our reply. We thank the reviewer for these suggestions. In the revised manuscript, we specified the quality of used chemicals and their exact molar ratio. We also cited two suggested papers in the introduction. We also extended the discussion part adding two last paragraphs into part 3.

  1. Authors are recommended to index PSXRD patterns with corresponding peaks. Laboratory XRD patterns can be presented as mentioned in the experimental condition. However, in Table 1 and the caption of Figure 3 crystal system (Tetragonal) can be mentioned. Moreover, Figure 3 is suggested to show the inset of the SXRD pattern (Figure 2). Figures 2 and 3 can be displayed and described at the begging of the results and discussion section.

Our reply: We thank the reviewer for these suggestions. In the revised manuscript, we included a new Figure 2 with a laboratory X-ray diffraction pattern and indexing results. A figure with the crystal structure (Figure 3 in the previous version) was moved to the inset of a new Figure 3 (Figure 2 in the previous version) as the reviewer suggested. The revised Table 1 includes more information (such as crystal system as the reviewer suggested).

  1. Why was CuO impurity included instead of other elements/compositions, although the authors used a stoichiometric ratio of raw materials? Please explain the fact. Authors can measure and show chemical analysis data of the final product to illustrate nonstoichiometric elements/compositions if any.

Our reply: We thank the reviewer for these suggestions. As the reviewer suggested, we added results of the EDX analysis of the chemical composition in the revised manuscript. The presence of CuO impurity suggests that the main phase should be slightly Cu-deficient as it was also confirmed by the EDX analysis.

  1. The authors are suggested to show the largest and the smallest particles on SEM image for better understanding.

Our reply: As the reviewer suggested, we showed an additional SEM image on the revised Figure 1 with a better magnification.

  1. Several typos, symbols presentation, spacing among words and units, commas, and full stops missing as well as several language corrections are recommended throughout the manuscript.

Our reply: We corrected the noticed typos.

Reviewer 2 Report

In this manuscript, the authors investigate the effects of Cu-doping on the B site of perovskite Sm2MnMn(MnTi3)O12. The long-range ferromagnetic ordering in the parent compound has been suppressed by replacing Mn2+ with Cu2+ while spin-glass state was observed. The behavior of spin-glass state was well characterized and the whole manuscript was very well written. I would recommend publication if the authors can address the following concerns:

1.    The relaxation curve shown in figure 8 should be inversed. Either the equation should be modified to M(0)-M(t) or the curves should be changed to negative region.

2.    In Figure 4, for ZFC curve under 100 Oe, there is an anomaly below 4 K. Can the authors briefly explain such behavior in the context?

Author Response

Reviewer 2.

  1. The relaxation curve shown in figure 8 should be inversed. Either the equation should be modified to M(0)-M(t) or the curves should be changed to negative region.

Our reply: The relaxation curves can be measured using two procedures. The first procedure includes cooling a sample in an applied magnetic field, setting the applied field to zero at the measurement temperature, and measurements of magnetization as a function of time. In this case, the magnetization usually decreases with time. This is what the reviewer wants to see on such a figure. The second procedure includes cooling a sample in zero magnetic field to the measurement temperature, setting an applied magnetic field, and measurements of magnetization as a function of time. In this case, the magnetization usually increases with time as shown on Figure 8 of our paper. As both procedures are widely used and reported in the literature, we would like to keep Figure 8 in its present form.

  1. In Figure 4, for ZFC curve under 100 Oe, there is an anomaly below 4 K. Can the authors briefly explain such behavior in the context?

Our reply: An anomaly on the ZFC 100 Oe curve near 4 K is extremely small (if the reviewer meant something below 4 K). A peak-like anomaly at 6 K (that is, at TSG) on the ZFC 100 Oe curve is a usual feature of a spin-glass transition as we discussed in the paper.

Reviewer 3 Report

General comment:

The authors studied the effects of Cu2+ doping on magnetic and dielectric properties of the parent Sm2MnMn(MnTi3)O12, which were prepared by a high-pressure, high-temperature method. This work is interesting and well described. I would like to recommend this manuscript to be published after the comments below addressed.

Comment 1:

Is there any possible application for the materials with spin-glass magnetic properties? The authors may mention it in the introduction.

Comment 2:

The authors should mentioned more synthesized details related to Sm2CuMn(MnTi3)O12 in experimental section.

Comment 3:

The authors mentioned a small amount of CuO impurity and the presence of Pt impurity in as-synthesized Sm2CuMn(MnTi3)O12. Is it possible to affect the dielectric and spin-glass magnetic properties of Sm2CuMn(MnTi3)O12? It is recommended to have some related discussion.

Comment 4:

There should be many coordinates for 4d, 4c, 2b, 8e, 8g, 8f. Why did the authors only mention one of coordinates for each of them?

Author Response

Reviewer 3.

  1. Is there any possible application for the materials with spin-glass magnetic properties? The authors may mention it in the introduction.

Our reply. As far as we know, there are no practical applications for the materials with spin-glass magnetic properties. Nevertheless, spin-glasses have attracted a lot of attention as described in many papers and one famous book (Ref. 30) as randomness, frustration, and glassiness represent very important phenomena in physics, and there is a richness of analogies with other fields. But as it is well described in the literature, we do not want to repeat such descriptions in our paper.

  1. The authors should mentioned more synthesized details related to Sm2CuMn(MnTi3)O12 in experimental section.

Our reply. Additional details of the synthesis procedure were mentioned as “Stoichiometric amounts of Sm2O3 (99.9 %), CuO (99.9 %), MnO (99.99 %), and TiO2 (99.9 %) were used as an initial oxide mixture with the 1:1:2:3 ratio, respectively. Commercial Sm2O3, CuO, and TiO2 chemicals were used. A single-phase MnO oxide was prepared from a commercial MnO2 chemical by annealing at 1273 K for 4 h in a 20% H2 + 80% Ar gas flow.”

  1. The authors mentioned a small amount of CuO impurity and the presence of Pt impurity in as-synthesized Sm2CuMn(MnTi3)O12. Is it possible to affect the dielectric and spin-glass magnetic properties of Sm2CuMn(MnTi3)O12? It is recommended to have some related discussion.

Our reply. In the revised manuscript, we added the following discussion. “Pt impurity was only observed in a powder sample, which could contain parts from the surface. The surfaces of a pellet used for dielectric measurements were polished. Therefore, Pt impurity should not present in a pellet and affect dielectric measurements.” “We note that CuO in an antiferromagnet with transition temperatures of 213 K and 230 K. Therefore, CuO impurity should not affect the reported magnetic properties at low temperatures.”

  1. There should be many coordinates for 4d, 4c, 2b, 8e, 8g, 8f. Why did the authors only mention one of coordinates for each of them?

Our reply. The note of Table 1 gives all the details. For example, the 8e site of space group 137 (origin choice 2) has all fixed (0, 0, 0) coordinates. Therefore, Table 1 does not list the coordinates for the 8e site. In case of the 8f site (x, -x, 1/4), only one coordinate x should be refined, and this refined coordinate is listed in Table 1.